# Comparison of Intensity- and Polarization-based Contrast in Amyloid-beta Plaques as Observed by Optical Coherence Tomography

Johanna Gesperger [1,2,*], Antonia Lichtenegger [1], Thomas Roetzer [2], Marco Augustin [1], Danielle J. Harper [1], Pablo Eugui [1], Conrad W. Merkle [1], Christoph K. Hitzenberger [1], Adelheid Woehrer [2,*] and Bernhard Baumann [1]

[1] Center for Medical Physics and Biomedical Engineering, Medical University of Vienna, 1090 Vienna, Austria; antonia.lichtenegger@meduniwien.ac.at (A.L.); marco.augustin@meduniwien.ac.at (M.A.); danielle.harper@meduniwien.ac.at (D.J.H.); pablo.euguiarrizabalaga@meduniwien.ac.at (P.E.); conrad.merkle@meduniwien.ac.at (C.W.M.); christoph.hitzenberger@meduniwien.ac.at (C.K.H.); bernhard.baumann@meduniwien.ac.at (B.B.)

[2] Institute of Neurology, General Hospital and Medical University of Vienna, 1090 Vienna, Austria; thomas.roetzer@meduniwien.ac.at

[*] Correspondence: johanna.gesperger@meduniwien.ac.at (J.G.); adelheid.woehrer@meduniwien.ac.at (A.W.); Tel.: +43-1-40400-39222 (J.G.); +43-1-40400-55040 (A.W.)



**Featured Application: The accumulation of pathological amyloid-beta protein is a key hallmark of Alzheimer's disease. In this study, we evaluated intensity- and polarization-sensitive optical coherence tomography as complementary methods for imaging amyloid-beta plaques as compared to quantitative pathology.**

**Abstract:** One key hallmark of Alzheimer's disease (AD) is the accumulation of extracellular amyloid-beta protein in cortical regions of the brain. For a definitive diagnosis of AD, post-mortem histological analysis, including sectioning and staining of different brain regions, is required. Here, we present optical coherence tomography (OCT) as a tissue-preserving imaging modality for the visualization of amyloid-beta plaques and compare their contrast in intensity- and polarization-sensitive (PS) OCT. Human brain samples of eleven patients diagnosed with AD were imaged. Three-dimensional PS-OCT datasets were acquired and plaques were manually segmented in 500 intensity and retardation cross-sections per patient using the freely available ITK-SNAP software. The image contrast of plaques was quantified. Histological staining of tissue sections from the same specimens was performed to compare OCT findings against the gold standard. Furthermore, the distribution of plaques was evaluated for intensity-based OCT, PS-OCT and the corresponding histological amyloid-beta staining. Only 5% of plaques were visible in both intensity and retardation segmentations, suggesting that different types of plaques may be visualized by the two OCT contrast channels. Our results indicate that multicontrast OCT imaging might be a promising approach for a tissue-preserving visualization of amyloid-beta plaques in AD.

**Keywords:** Alzheimer's disease; histology; microscopy; neuroimaging; polarization

---

## 1. Introduction

Alzheimer's disease (AD) is the leading cause of dementia accounting for up to 70% of cases [1]. In 2015, up to 47 million people were affected worldwide [2–4]. This number is expected to double every 20 years, resulting in a dramatically increased social as well as financial burden [5]. AD is characterized

by the degradation of neurons, especially in the cerebral cortex, basal forebrain, hippocampus and amygdala, critical regions for memory and learning as well as cognitive functioning [6–12]. The disease manifests with mild cognitive deficits progressing to significant cognitive impairments over time [13]. As the disease progresses, patients become increasingly dependent on caregivers [14]. Current research focuses on developing targeted therapies for the disease and improving diagnostics of AD, for instance by investigating imaging biomarkers [15].

Key hallmarks of AD are the deposition of intracellular neurofibrillary tangles and extracellular accumulation of pathological amyloid-beta protein [16]. The gold standard for the definitive diagnosis of AD is the histological analysis of different brain regions including multiple cortical regions [17]. For this purpose, the tissue has to be fixed in formalin, dehydrated and embedded in paraffin. Using immunohistochemistry, thin sections of the tissue are then stained in order to determine the presence or absence of amyloid-beta protein in the brain [18]. Usually, a further section is stained with Congo red dye to visualize a subclass of amyloid-beta plaques, so-called neuritic plaques. Due to the parallel arrangement of the amyloid fibrils within them, neuritic plaques exhibit the physical phenomenon known as birefringence [19]. This birefringence is amplified by Congo red and can therefore be visualized with polarized light microscopy [20]. However, the histological procedure requires formalin fixation, which terminates cell metabolism, does not preserve lipid structures such as cell membranes, and shrinks the tissue [21,22]. An alternative to conventional histology is the sectioning of fresh frozen tissue. This method does not require any of the previously mentioned tissue preparation procedures, which makes it a rapid and powerful approach for intraoperative diagnostics. However, studies have shown that the frozen-section procedure is not as accurate as conventional formalin-fixed and paraffin-embedded histological analysis of tissue and that pathologists often have to deal with artifacts due to the rapid freezing process [23,24].

Optical coherence tomography (OCT) is an optical imaging technique, which was introduced in the early 1990s [25] and has since become a standard diagnostic tool in ophthalmology [26]. OCT has also been used in many other fields including dermatology, endoscopy and neuroimaging [27–29]. OCT data can be acquired in three-dimensional (3D), in real time, and with micrometer resolution [25]. Most importantly, tissues can be imaged non-destructively in an imaging range of a few millimeters. OCT image contrast is based on the inherent backscattering properties of the sample, so that extensive tissue preparation such as fixation is not required. A functional extension of OCT is polarization-sensitive (PS) OCT [19,30]. PS-OCT visualizes specific tissue types based on how they alter the polarization state of the incident light beam. For example, birefringence exhibited by fibrous tissues and depolarization produced by melanin pigments are sources of PS-OCT image contrast and can be assessed by PS-OCT [25,31].

Several recent reports have demonstrated that amyloid-beta plaques in both human and mouse brain tissue can be visualized using either conventional OCT, PS-OCT [19,32–36] or other functional extensions such as spectroscopic OCT [32] or directional OCT [36]. Label-free imaging of cerebral amyloid-beta plaques has been achieved using extended-focus optical coherence microscopy (OCM) in the near infrared range at 800 nm [34]. By shifting to the visible spectrum to achieve sub-micrometer axial resolution, extended-focus OCM has been used to investigate plaques and small fiber tracts in ex-vivo cerebral mouse tissue [35]. A PS-OCM setup in the near infrared around 800 nm enabled the visualization of neuritic amyloid-beta plaques in brain tissue of human AD patients. These measurements provided information about the polarization characteristics of these pathological structures [19]. Finally, a spectral-domain visible light OCM system with sub-micrometer axial resolution has been used to investigate the inherent scattering contrast and the spectroscopic properties of amyloid-beta plaques in ex-vivo human and mouse brain tissue [32,33]. To the best of our knowledge, no previous study has evaluated the capabilities of OCT and PS-OCT for amyloid-beta imaging in a large human sample cohort. In this article, we present a comparison of conventional OCT and PS-OCT for the visualization of amyloid-beta plaques in human brain tissue of 11 patients. We relate the plaque

load observed in both of the OCT contrast channels to that revealed by quantitative histology and evaluate the advantages of multi-contrast imaging.

## 2. Materials and Methods

### 2.1. Brain Samples

Formalin-fixed post mortem human brain samples (5 mm × 5 mm × 5 mm) from 11 patients diagnosed with AD (Braak stages III to VI, Thal phases 2–5) were evaluated in this study. In total, 20 data sets were acquired. The patient cohort consisted of five females and six males aged between 63 and 89 years. All brains were provided by the Neurobiobank of the Medical University of Vienna, Austria (ethics approval number 396-2011). Cortical regions of each patient were investigated by PS-OCT and histology.

### 2.2. OCT and Data Acquisition

A spectral-domain PS-OCT system (TEL220PSC2, Thorlabs, Lübeck, Germany) was utilized for imaging the brain samples. The setup used a multiplexed superluminescent diode in the near infrared (central wavelength 1300 nm, bandwidth 170 nm) and achieved a theoretical axial resolution of 5.5 μm in air. The theoretical lateral resolution was 7 μm (at the focal plane), using a commercial scanning lens with a focal length of 18 mm (OCT-LK2, Thorlabs, Lübeck, Germany). The focus was positioned at the tissue surface for imaging. For the acquisition, a lateral (x,y) field of view of 500 pixels × 500 pixels in a cortical region of the sample was imaged, corresponding to 500 μm × 500 μm. The imaging depth range was 3.5 mm in air. The A-scan rate was 76 kHz and a sensitivity of 109 dB was measured. OCT images displaying backscattered intensity and retardation were computed from the raw spectral data. All volumes were saved as image stacks for further processing.

### 2.3. Histology

After OCT imaging, the whole fixed AD cerebral tissue samples were dehydrated and embedded in paraffin in preparation for histopathological workup. The brain samples were cut into 2.5 μm thick sections on a microtome. One slice was stained immunohistochemically using an anti-amyloid-beta antibody (clone 6F/3D, diluted 1:100, Dako, Santa Clara, CA, United States). With a second tissue slice, Congo red (CI22120, 0.5% dissolved in 50% ethanol, Merck, Kenilworth, NJ, United States) staining was performed to visualize neuritic plaques. Sections stained by Congo red were imaged using both bright-field and polarized light microscopy (BH2-RFCA, Olympus, Hamburg, Germany). In addition, for one specimen, 180 consecutive 3 μm thick serial sections were cut and stained immunohistochemically against amyloid-beta. All sections were further scanned using a slide scanner (NanoZoomer 2.0 HT, Hamamatsu, Shizuoka, Japan). Afterwards, the Fiji [37] plugin StackReg [38] was used to align the serial micrographs in order to generate a three-dimensional (3D) rendering of the stained AD cerebral tissue.

### 2.4. Data Processing and Statistical Analysis

Using ITK-SNAP, the plaques were manually segmented in the intensity and the retardation data [39]. The resulting volumetric segmentation data were saved as binary files. Using the 3D Objects Counter plugin [40] in Fiji and the segmentation data of the plaques, their diameter, area and plaque density (in plaques per $mm^3$) were quantified over the whole volume [37]. The mean plaque diameter in μm and plaque density were also extracted by the 3D Objects Counter tool. For plaque diameters, the average of the horizontal and vertical diameter was calculated.

The mean intensity and retardation signals ($\mu$) of OCT image data were calculated for the plaques ($\mu_P$) and the surrounding brain parenchyma ($\mu_{BP}$) using the retardation averaging method 1 [41]. To estimate the signal levels in the brain parenchyma, the same amount of pixels used for the evaluation of the signal in the plaques was used. Values adjacent to the plaques at the same depth (located 10 μm to the left in OCT tomograms) were chosen. Using a custom Matlab (Version R2015b, The MathWorks, Inc., Natick, MA, United States) code, the Weber contrast (WC) [42] was calculated in both contrast modalities for the plaques compared to the surrounding cortical gray matter:

$$WC = \frac{\mu_P - \mu_{BP}}{\mu_{BP}} \tag{1}$$

WC was evaluated for the intensity data (Int) and for the retardation data (Ret) in linear scale based on the plaque segmentations performed in both the intensity data ($Int_{Seg}$) and the retardation data ($Ret_{Seg}$), resulting in four different WC analyses. The same evaluation was performed for the signal to noise ratio (SNR), where $\sigma_P$ and $\sigma_{BP}$ are the standard deviation of the signals in plaques and the brain parenchyma, respectively [42]:

$$SNR = \frac{\mu_P - \mu_{BP}}{\sqrt{\sigma_P^2 + \sigma_{BP}^2}} \tag{2}$$

Furthermore, the plaque overlap (i.e., the percentage of plaques segmented in both intensity and retardation images) was evaluated using Fiji. Finally, histograms and box plots were generated for visual representation. Student's t-tests were used to analyze statistical differences for equal mean values and Bonferroni correction was applied when evaluations on multiple groups were performed. The data are presented with their associated standard deviations, unless otherwise stated.

For a direct comparison to histology, the number of amyloid-beta plaques in fields of view (FOV) of 500 μm × 500 μm × 2.5 μm (corresponding to the thickness of a histological section) were quantified for OCT intensity images, PS-OCT retardation images and the corresponding histological section of one specimen. The dense neuritic and diffuse plaques in one histological slice were manually counted in five randomly chosen regions and the mean plaque diameter and the areal plaque density (in plaques per mm$^2$) were evaluated.

## 3. Results

### 3.1. Appearance of Plaques in Intensity and Retardation OCT Images

Samples from eleven brains affected by AD were imaged. Figure 1a,b show the intensity en-face projection over a depth of 10 μm and a representative B-scan, respectively. Figure 1d,e show the corresponding retardation en-face projection and B-scan. Plaques appeared as highly scattering features in the intensity OCT images. In retardation data sets, plaques exhibited increased phase retardation compared to the polarization preserving brain parenchyma. As cumulative phase retardation effects occur, some plaques exhibit retardation trails. This effect can be observed in Figure 1e in the plaque on the right. Plaques visible in both OCT and PS-OCT data are highlighted with yellow arrows. Reconstructions of the acquired 3D volumes are displayed in Figure 1c,f.

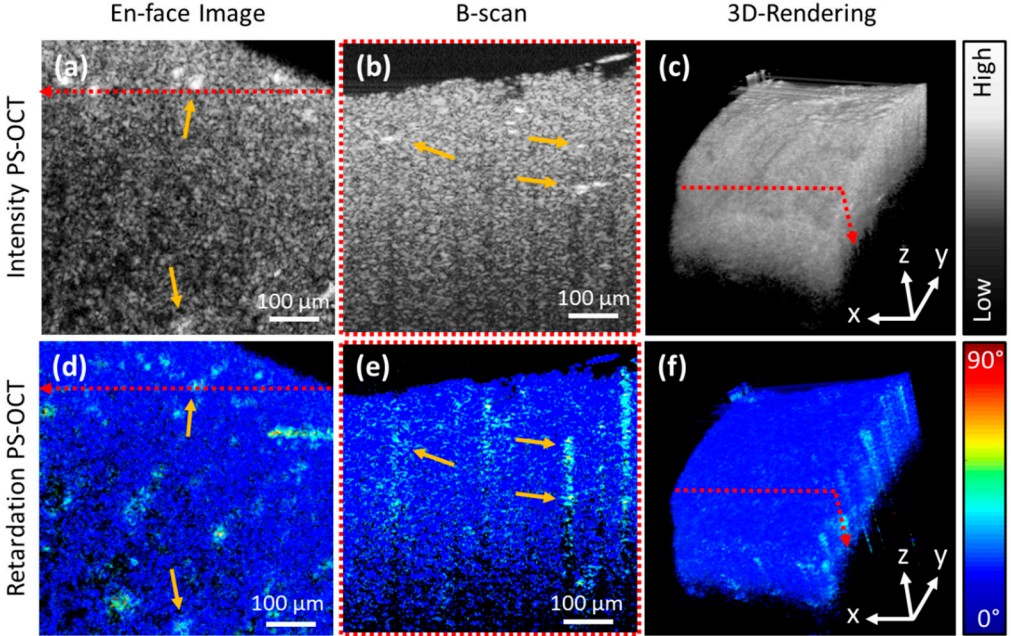

**Figure 1.** Visualization of amyloid-beta plaques in intensity and retardation optical coherence tomography (OCT) images. (**a**) Intensity en-face projection image averaged over a depth of 10 μm starting 50 μm underneath the tissue surface. (**b**) Representative intensity B-scan image. (**c**) Three-dimensional (3D) volume of intensity data. (**d**) Retardation en-face projection image averaged over 10 μm starting 50 μm underneath the tissue surface. (**e**) Representative retardation B-scan image. (**f**) 3D volume of retardation data.

## 3.2. Comparison of Segmentation and Image Contrast

Figure 2a,b show en-face projection images over 50 μm for intensity and retardation data, respectively. Hyperscattering or turquoise pixels indicate amyloid-beta plaques. B-scans with manually segmented plaques highlighted as red and yellow dots are displayed in Figure 2c,d. Representative 3D renderings of the segmentations based on the intensity and phase retardation images are shown in Figure 2e,f, respectively. In Figure 2h, an overlay of both segmentation data sets is shown. Only few plaques appear in both data sets. The amount of plaques overlapping in both data sets was quantified as shown in Figure 2g. On average, 5% of the amyloid-beta plaques in all data sets were visualized by both contrast channels.

Weber contrast was assessed in the intensity and retardation data sets based on the segmentation of both the intensity data ($Int_{Seg}$) and retardation data ($Ret_{Seg}$). As shown in Figure 3a, the image contrast of plaques was higher in intensity data and $Int_{Seg}$ compared to intensity data and $Ret_{Seg}$; however, no significance was found ($p = 0.8$). The same analysis for the retardation values showed a significant contrast difference when comparing plaques segmented for retardation and intensity data ($p = 0.02$), see Figure 3b. Additionally, the SNR was assessed in the intensity and retardation data sets based on the segmentation of both the intensity data ($Int_{Seg}$) and retardation data ($Ret_{Seg}$). A slightly higher difference for intensity images and $Int_{Seg}$ was observed compared to intensity images and $Ret_{Seg}$ ($p = 0.6$). For retardation data and $Ret_{Seg}$, a higher SNR was calculated when compared to retardation images and $Int_{Seg}$ ($p = 0.06$). A detailed overview of plaques segmented in all data sets is shown in Figure 4a.

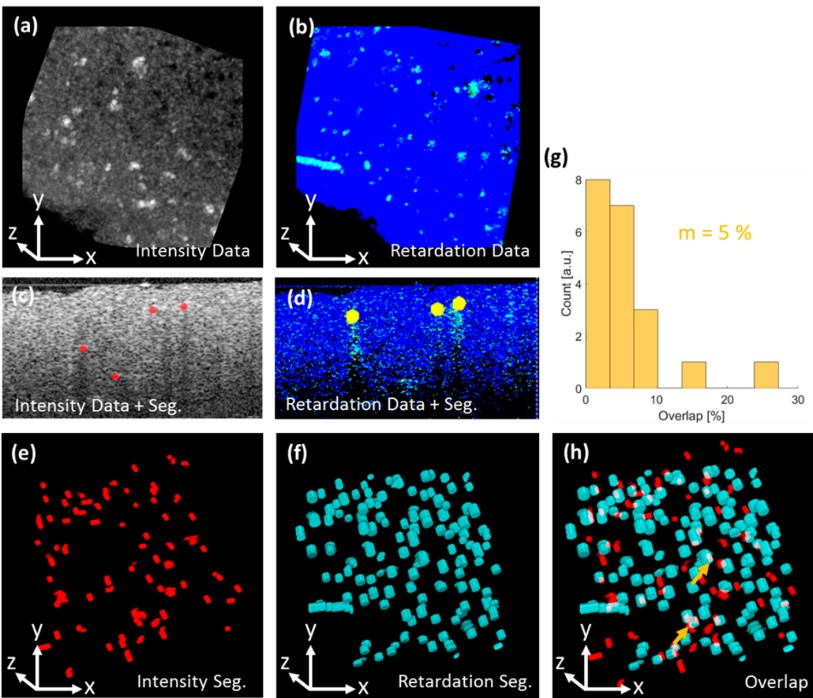

**Figure 2.** Comparison of plaque segmentation in intensity and retardation data. (**a**) Maximum intensity en-face projection image. (**b**) Median retardation value en-face projection over 50 μm. Color map: see Figure 1. (**c**) Intensity B-scan with segmented plaques shown in red. (**d**) Retardation B-scan with segmented plaques shown in yellow. (**e**) Representative 3D rendering of the segmented plaques in an intensity volume. (**f**) Representative 3D rendering of the segmented plaques in a retardation data set. (**g**) Quantification of total plaque overlap in every OCT and PS-OCT data set, i.e., plaques segmented in both contrast channels (m indicates the mean overlap). (**h**) Volumetric overlay of the intensity and retardation segmentations. Two yellow arrows highlight two plaques visible both in intensity and retardation image data. White pixels indicate plaques which are visible in both intensity and retardation data.

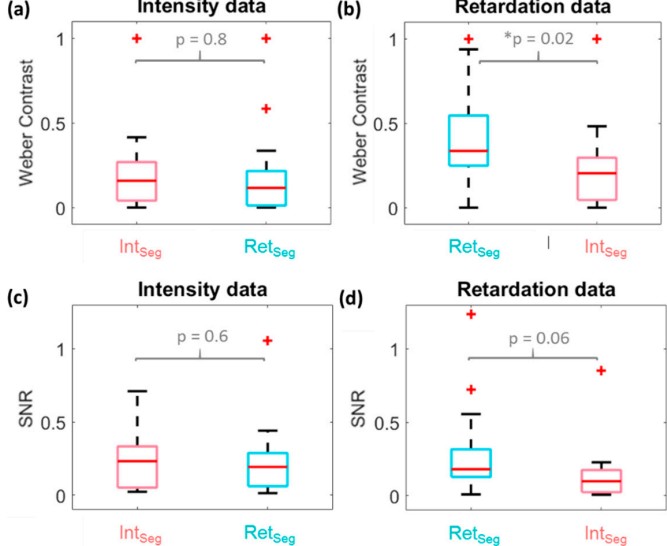

**Figure 3.** (**a**) The Weber contrast (WC) was calculated for all intensity data sets in linear scale. WC for the plaques segmented in the intensity data (Int$_{Seg}$) compared to those segmented in the retardation data (Ret$_{Seg}$) was evaluated. A slightly higher but not statistically significant difference for intensity data and Int$_{Seg}$ was observed compared to intensity data and Ret$_{Seg}$. (**b**) WC for the retardation data.

A significant WC difference for retardation data and Ret$_{Seg}$ over retardation data and Int$_{Seg}$ was calculated (*p*-value 0.02). (**c**) The signal to noise ratio (SNR) was calculated for all intensity data sets in linear scale. SNR for the plaques segmented in the intensity data (Int$_{Seg}$) compared to those segmented in the retardation data (Ret$_{Seg}$) was evaluated. A slightly higher but not statistically significant difference for intensity data and Int$_{Seg}$ was observed compared to intensity data and Ret$_{Seg}$. (**d**) The SNR for the retardation data. A slightly higher but not statistically significant difference for retardation data and Ret$_{Seg}$ was observed compared to retardation data and Int$_{Seg}$.

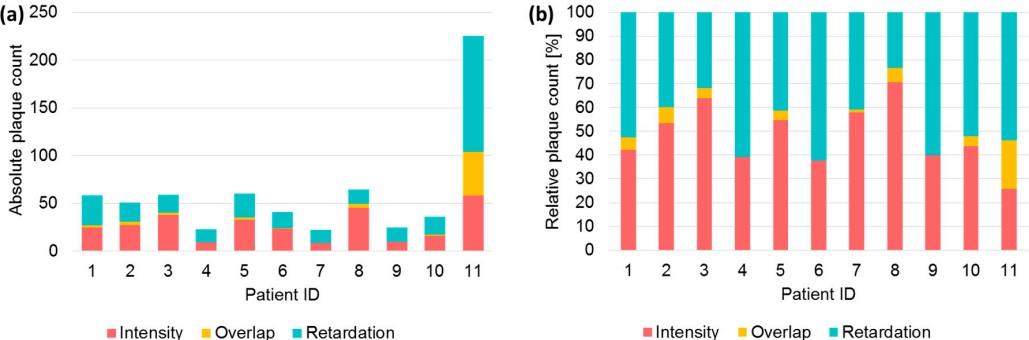

**Figure 4.** Overview of plaques segmented in the OCT data sets of eleven patients in 500 μm × 500 μm × 500 μm fields of view (FOV). (**a**) Plaque count in retardation and intensity data including the overlapping plaques (i.e., plaques observed both in intensity and retardation images) plotted for each patient. (**b**) Relative plaque count in retardation and intensity data including the overlapping plaques plotted with age. Patients with two data sets were averaged and mean values are plotted.

### 3.3. Evaluation of Plaque Load and Size Using Multicontrast OCT

The diameters of the plaques segmented in intensity and retardation data was evaluated. The size distributions of all data sets revealed by the two OCT contrast channels are shown in the histogram in Figure 5a. In the intensity images, the diameters ranged from 11 to 85 μm with a mean value of 42 μm. The plaque size observed in the retardation data ranged from 14 to 148 μm in diameter (mean 54 μm). Next, the observed plaque volume density (plaques per mm$^3$) was analyzed and is plotted in Figure 5b. The plaque density in intensity data (median of 224 [Q1–Q3: 112–332 p/mm$^3$]) and retardation data (median of 156 [Q1–Q3: 124–208 p/mm$^3$]) was rather similar. Lastly, the mean plaque area (in μm$^2$) was calculated for each data set. Results are displayed in box plots in Figure 5c. In intensity data, plaque areas ranged between 1250 and 1450 μm$^2$ (median of 1394 [Q1–Q3: 1256–1458 μm$^2$]). This is significantly smaller than those measured in retardation images, where the plaque area was clustered between 1400 and 1750 μm$^2$ (median of 1518 [Q1–Q3: 1430–1727 μm$^2$]). Q1 and Q3 represent the first and third quartile.

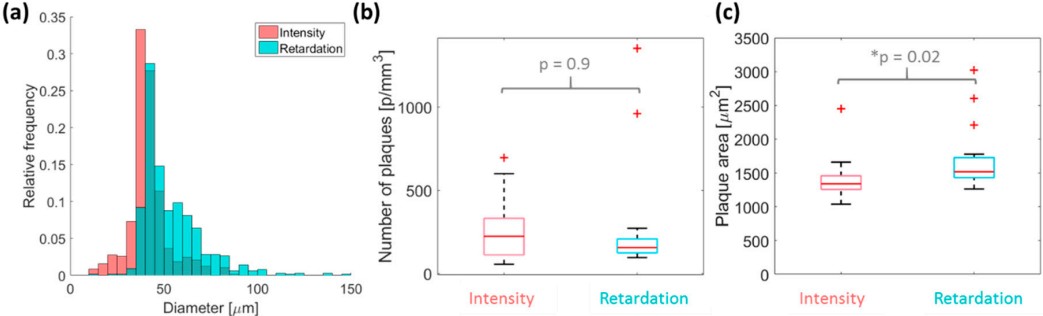

**Figure 5.** Statistical evaluation of segmented data sets. (**a**) Histogram of plaque diameters measured for intensity and retardation data sets. (**b**) Absolute numbers of plaques per mm$^3$ in intensity and retardation images. The amount of plaques visualized in intensity data (median of 224 [Q1–Q3: 112–332 p/mm$^3$]), as compared to retardation data (median of 156 [Q1–Q3: 124–208 p/mm$^3$]) was rather similar. (**c**) Mean plaque size in μm$^2$ calculated for intensity (mean 1394 ± 294 μm$^2$) and retardation (mean 1750 ± 608 μm$^2$) data. For boxplots, mean values of each data set were computed.

### 3.4. Comparison to Histology

Immunohistochemical stainings against amyloid-beta were performed in order to confirm OCT findings. In Figure 6a, a 3D reconstruction and rendering of 180 stained serial sections is shown. Figure 6b displays a zoom-in into the cortical region of the brain sample. Immunohistochemically stained amyloid-beta plaques appear as brownish deposits. Neuritic plaques appear as red accumulations when labeled with Congo red and were investigated with brightfield microscopy (see Figure 6c). This staining was performed for each brain sample as well. Evaluating the same Congo red stained section using polarization-contrast microscopy, neuritic plaques exhibit an apple-green color with a characteristic cross shape as displayed in Figure 6d.

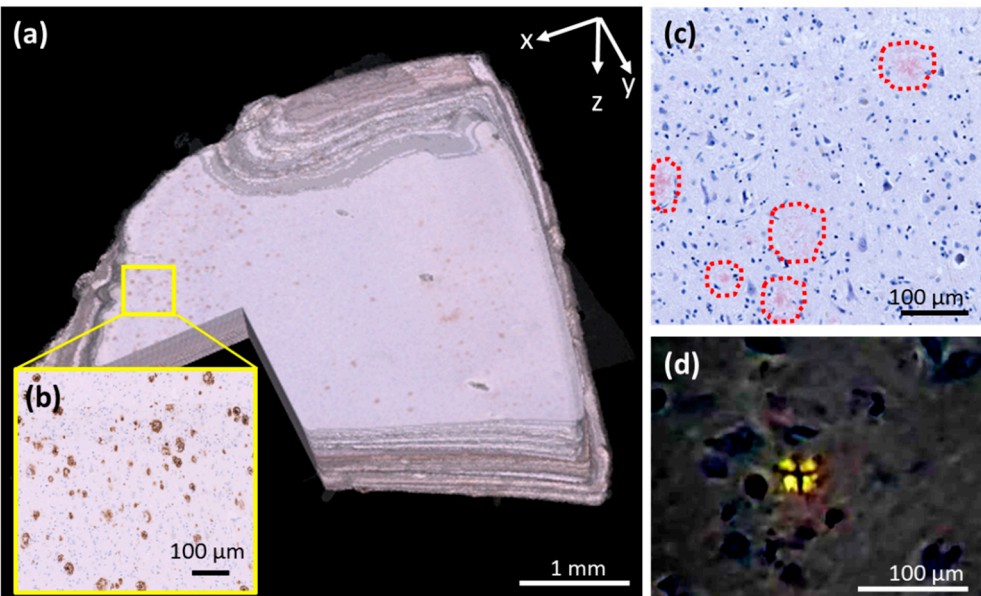

**Figure 6.** Histological analysis of Alzheimer's disease (AD) human brain samples. (**a**) Three-dimensional reconstruction and rendering of 180 immunohistochemically stained serial sections. (**b**) Zoom-in into a plaque-rich cortical region where brown staining indicates amyloid-beta deposits. (**c**) Congo red stained section imaged by bright-field microscopy. Plaques appear as reddish accumulations. Red circles indicate neuritic plaques. (**d**) Congo red staining investigated by polarized light microscopy. A neuritic plaque can be observed in characteristic apple-green color and cross-shaped form.

For a direct comparison of OCT and state-of-the-art histology, the number of amyloid-beta plaques was evaluated for intensity-based OCT, PS-OCT and the corresponding histological amyloid-beta stainings in five regions of one patient. In the immunohistochemically stained histological section, $240 \pm 64$ plaques/mm$^2$ were counted. Of these plaques, 33.5% ($\hat{=} 79.2 \pm 20.8$ plaques/mm$^2$) were classified as cored/dense and 66.5% ($\hat{=} 160.8 \pm 42.8$ plaques/mm$^2$) as diffuse. In the corresponding OCT intensity and retardation images, plaque densities of $34 \pm 10$ plaques/mm$^2$ and $148 \pm 32$ plaques/mm$^2$ were measured, as shown in Figure 7a. The diameters of the diffuse and dense plaques annotated in histological sections were evaluated and plotted in the histogram shown in Figure 7b. Representative images of a dense plaque (diameters ranged from 4.8 µm to 25.5 µm) and a diffuse plaque (diameters ranged from 6.9 µm to 83.6 µm) are shown in Figure 7c,d, respectively. The histology data showed that significantly fewer plaques are visualized by OCT ($p < 0.01$), and they appeared significantly bigger ($p < 0.01$).

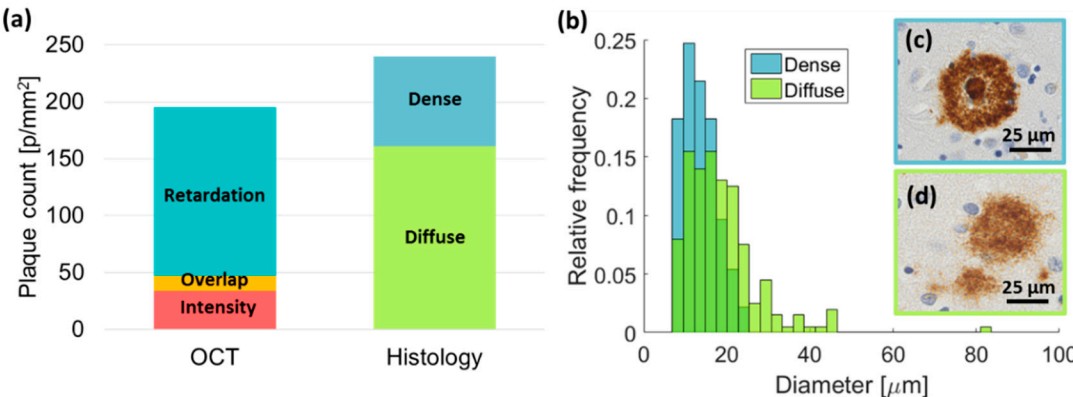

**Figure 7.** Comparison of histology and OCT data. (**a**) Absolute plaque count (p/mm$^2$) in retardation and intensity data including those which overlap, compared to diffuse and dense plaques in histology. (**b**) Histogram of the plaque diameter in dense (4.8–25.5 µm) and diffuse plaques (6.9–83.6 µm) evaluated in histological slices. (**c**) A typical image of a dense plaque. (d) A typical image of a diffuse plaque.

## 4. Discussion

The post-mortem assessment of amyloid-beta plaques is an essential component of neuro-pathological AD diagnostics [16]. In this article, we investigated the imaging capabilities of OCT intensity and polarization contrast as alternative techniques for the visualization and quantitative assessment of amyloid-beta plaques in ex-vivo cerebral tissue of AD patients.

PS-OCT visualizes amyloid-beta plaques based on their optical scattering and birefringent properties [33–36]. Previous studies mainly exploited the hyperscattering characteristics of plaques and only one report investigated the birefringent characteristics of neuritic plaques using PS-OCT [19]. Here, we presented a comparative analysis of the plaque visualization capabilities of intensity- and PS-OCT contrast. Surprisingly, most plaques appeared in only one contrast channel and were in most of the cases not visible in the other. While for each patient, we observed different fractions of plaques in intensity and retardation respectively (Figure 4), only very few plaques—on average 5%—did actually overlap (Figure 2). This finding may indicate different underlying contrast mechanisms, suggesting that distinct subgroups of plaques provide independent optical contrast. Figure 7 may suggest that intensity and retardation contrasts visualize different plaque morphologies. However, further investigation and a one-on-one correlation to histology would be required in order to elucidate the relationship between the two OCT contrast channels with the different subgroups of plaques.

In our earlier work, we simulated the birefringent [19] and scattering signals [36] produced by neuritic plaques assuming a geometry of radiating fibrils [43]. These simulations suggested that neuritic plaques are birefringent, resulting in increased phase retardation observed in PS-OCT [19]. As cumulative phase retardation effects occur, some plaques exhibit retardation trails. In our recently published simulations we found that low numerical aperture (NA) values resulted in straight retardation shadows in z-direction whereas higher NA values produced a lateral spread of those shadows [19]. The intensity of directly backscattered light was found to be rather low for this plaque geometry, such that strong neuritic plaque signals were only observed using a dark-field detection scheme for OCT [36].

By analyzing the signal levels in plaques compared to the surrounding gray matter, most birefringent plaques (i.e., those visualized in retardation images) were invisible in co-localized OCT intensity data. However, higher retardation was observed only in these birefringent plaques as compared to those highlighted in the intensity images. This finding, which is also supported by previous research reporting the visualization of plaques by OCT without polarization contrast [31–35], reinforces our assumption that intensity and retardation-based contrasts are caused by different families of amyloid-beta plaques. To further evaluate this, WC and SNR calculations were performed. As Figure 3 shows, there was no significant difference either in WC or SNR when comparing intensity

data. This slight difference may be explained by a depth-dependent contrast increase between plaques and surrounding brain parenchyma. However, note that plaques vary not only in size but also in shape and density. Therefore, in intensity images, many plaques appeared as strong hyperreflective spots and only some expressed a vague hyperreflectivity, resulting in weaker WC and SNR values. For retardation images, WC differences were significant. These findings indicate that plaques in retardation images overall express a higher contrast than plaques visualized in intensity data when compared to surrounding neural tissue.

The plaque sizes as measured by histology and the two different OCT contrasts differed from each other. The diameters of the plaques segmented in the retardation images showed a trend to be bigger than those segmented in the intensity data (Figure 5a). Plaques segmented in histology data, the gold standard, had a smaller plaque diameter when compared to plaques segmented in retardation and intensity images (Figure 5a versus Figure 7b). The minimum plaque size detected in histology was around 3 µm, whereas the smallest plaques detected in OCT were around 12 µm. While the detected size of plaques in all modalities did agree with earlier literature reports [44–46], the direct comparison between histology and OCT (Figures 5a and 7) showed that OCT seems to severely overestimate plaque size. Potential reasons for this could be the expansion (convolution) of imaged objects by the imaging system's point spread function (PSF), the inherently different contrast mechanisms and tissue shrinkage due to dehydration and fixation processes. Previous studies reported tissue shrinkage due to histologic workup up to 17% [47,48]. Most importantly, the image resolution of OCT is rather poor when compared to histology. The observed size of a point-like object is described by the PSF of the imaging system. In an image, any object will be smeared out (convolved) with the PSF in x, y and z directions. The PSF in OCT is described by the coherence length in z-direction (which is determined by the covered spectral bandwidth in wavenumber space) and by the beam spot size in x-y direction [49]. Moreover, the beam spot size depends on the distance to the focal plane. Due to the limited resolution, multiple single plaques in close vicinity detected by histology might also be mistaken for a single, large plaque in OCT. In contrast, immunostaining visualizes amyloid-beta deposits of (almost) all shapes and sizes very specifically, and is expected to pick up much fainter signals and structures. OCT seems to visualize much fewer plaques (Figure 7a).

Several methodological modifications may improve the performance of OCT for visualizing amyloid-beta plaques. In order to enhance the axial image resolution, a light source providing a broader bandwidth in wavenumber space (e.g., with a shorter central wavelength and/or broader wavelength coverage than the system used here) could be employed [32,33]. Furthermore, objective lenses with higher magnification could be used for improving the lateral resolution. Note, however, that the use of objectives with a shorter focal length usually comes at the cost of a reduction of the FOV.

As a logical next step, tissues of patients with different stages of the disease could be investigated to evaluate the time point at which plaques can be visualized with OCT. Further elucidation in the three-dimensional shape could be obtained by reconstructed histological serial sections similar to that shown in Figure 6a.

As an advantage over histology, OCT can investigate ex-vivo cerebral tissue in a tissue-preserving manner, without any fixation, sectioning or staining. However, at the same time, OCT relies on intrinsic contrast and therefore does not require additional tissue processing steps. Hence, for some applications, multimodal OCT imaging might be a promising high throughput approach for the investigation of pathological alterations of the human brain.

## 5. Conclusions

A commercial polarization-sensitive optical coherence tomography (PS-OCT) setup was used to investigate ex-vivo human brain samples of patients diagnosed with Alzheimer's disease (AD). The performance of multicontrast OCT for the imaging of amyloid-beta plaques, one hallmark of the disease, was investigated. Following OCT imaging, histological staining was performed for a comparison to the gold-standard technique for AD diagnosis. Plaques were segmented in the OCT

intensity and retardation data sets and the plaque contrast was analyzed. Notably, complementary distinct plaque patterns were observed in both contrast channels, only 5% of plaques were visualized in both intensity- and retardation-based contrast. The segmentation data were further used to evaluate the plaque diameter, area and the number of plaques per mm$^3$. The same parameters were evaluated for histology data, showing that significantly fewer plaques are visualized by OCT and that they appeared significantly bigger ($p < 0.01$). In the OCT intensity contrast, a similar amount of plaques was counted compared to the retardation data. Plaques in retardation images appeared significantly larger. In conclusion, the results suggest that a multicontrast OCT is a promising tissue-preserving imaging technique to investigate amyloid-beta plaques.

**Author Contributions:** Conceptualization, B.B., A.L., J.G., C.W.M. and M.A.; C.K.H.; methodology, J.G., A.L., T.R. and A.W.; software, A.L., M.A. and P.E.; validation, J.G., A.L. and D.J.H.; formal analysis, J.G. and A.L.; investigation, J.G., A.L. and D.J.H.; resources, A.L., J.G. and C.K.H.; data curation, A.L., J.G., T.R. and P.E.; writing—original draft preparation, J.G., A.L. and B.B.; writing—review and editing, A.L, J.G., B.B., D.J.H., C.K.H., M.A., C.W.M., P.E. and T.R.; visualization, A.L. and J.G.; supervision, B.B. and A.W.; project administration, B.B.; funding acquisition, B.B.

**Funding:** This research was funded by the European Research Council Starting Grant 640396 OPTIMALZ.

**Acknowledgments:** The authors want to thank Thorlabs and their PS-OCT team for providing the PS-OCT setup and support during imaging and post-processing. We want to thank Ellen Gelpi for her expertise in neurodegeneration and the evaluation of histologic stainings and the whole neuropathology lab of the Medical University of Vienna. We particularly acknowledge Florian Beer for his support while setting up the OCT system and performing the measurements. This work was funded by the European Research Council (ERC StG 640396 OPTIMALZ).

**Conflicts of Interest:** The authors declare no conflict of interest.

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
