# Peer review of "Comparison of Intensity- and Polarization-based Contrast in Amyloid-beta Plaques as Observed by Optical Coherence Tomography"

_applsci, doi:10.3390/app9102100_

Reviewer 1 Report

Gesperger et al present a study on the use of spectral domain optical coherence tomography (OCT) and polarization sensitive OCT (PS-OCT) to detect amyloid-beta plaques in Alzheimer’s disease patient brain tissue slices. Furthermore, the authors compare their results to the standard immunohistochemical stainings of the same slices following OCT imaging. While the capability of OCT and PS-OCT to detect plaques have been demonstrated previously, this article aims at a rigorous study with multiple patient samples.

While the technique and data analysis appear to be quite robust, the article might gain from a more exhaustive one-on-one comparison of the OCT data with the histological analysis instead of the comparison of pooled data from all the patients.

The analysis revealed a significant increase in the size of the plaques in OCT and PS-OCT images. It would be interesting to note if this increase is linear for all the samples. Perhaps a scaling factor could be calculated.

Author Response

Please see enclosed pdf.-file for responses.

Reviewer 2 Report

Review

This is a manuscript that qualitatively and quantitatively compares the intensity and polarization contrast optical coherence tomography (OCT) for amyloid-beta plaque detection in the context of detection neurodegenerative disorders, e.g. Alzheimer’s disease (AD). While in vivo/in vitro microscopy via OCT is a powerful diagnostic methodology currently used in the best clinics worldwide, qualitative pathology (conventional sectioning histology) remains a gold standard. The comparison between intensity-, polarization-based and histology was presented. The authors reported that OCT is a powerful complementary modality and demonstrated promising results. I think the paper could prove to be interesting and useful to an audience, making it acceptable for publication in the Journal.

Minor critiques:

-        (Line 55) Reference regarding tissue shrinkage: Schulz, et al. “Three-Dimensional Strain Fields in Human Brain Resulting from Formalin Fixation.” Journal of Neuroscience Methods, vol. 202, no. 1, 2011, pp. 17–27.

-        (Line 47) Comparison to cryo-histology might be of interest to the readers.

-        The consistency of naming might be improved, e.g. PS-OST system (Thorlabs TEL220PSC2) vs. Hamamatsu NanoZommer 2.0 HT slide scanner.

-        Figure 1: consider removing.

-        Figure 2f: Consider improving contrast.

-        Figure 3: Consider adding ground truth data. While plagues visible in both image data are highlighted, it remains unclear which plagues visible only in one dataset is real. Please consider combining Figure 7 and Figure 3.

Author Response

Please see enclosed pdf.-file for Responses.

Reviewer 3 Report

This study compared the distribution of plaques in brain tissues from 11 patients of alzheimer’s disease visualized by the optical coherence tomography (OCT) and histology. The authors found that fewer plaques were visualized by OCT. Plaques in retardation images appeared significantly larger. The authors conclude that OCT is a promising tissue preserving imaging technique to investigate amyloid-beta plaques.

 1. Please specify how do you make sure that OCT image and histology image were acquired at the same region of the specimen.

 2. Page 11. Any tissue shrinkage after histology processing?

Author Response

(The authors gave the same response as above.)
